# Occlusal Disharmony—A Potential Factor Promoting Depression in a Rat Model

**DOI:** 10.3390/brainsci12060747

**Published:** 2022-06-07

**Authors:** Sihui Zhang, Ling Wu, Mi Zhang, Kaixun He, Xudong Wang, Yuxuan Lin, Shuxian Li, Jiang Chen

**Affiliations:** 1School and Hospital of Stomatology, Fujian Medical University, Fuzhou 350004, China; zhangsihui@fjmu.edu.cn (S.Z.); wuling4829@fjmu.edu.cn (L.W.); zhangmi@fjmu.edu.cn (M.Z.); drkesonho@fjmu.edu.cn (K.H.); linyuxuan@fjmu.edu.cn (Y.L.); 2Fujian Key Laboratory of Oral Diseases, School and Hospital of Stomatology, Fujian Medical University, Fuzhou 350004, China; wyudon963@163.com (X.W.); leel2020@126.com (S.L.)

**Keywords:** occlusal disharmony, chronic unpredictable stress, depression-like behavior, hypothalamus–pituitary–adrenal axis, 5-HT system

## Abstract

Objectives: Patients with occlusal disharmony (OD) may be susceptible to depression. The hypothalamus–pituitary–adrenal axis, 5-HT and 5HT_2A_R in the prefrontal cortex (PFC), amygdala, and hippocampus are involved in the modulation of emotion and depression. This study investigated whether OD affects the HPA axis and 5-HT system and, subsequently, produces depression-like behaviors in rats. Materials and methods: OD was produced by removing 0.5 and 0.25 mm of hard tissue from the cusps of the maxillary molars in randomly selected sides of Sprague–Dawley rats. CUS involved exposure to 2 different stressors per day for 35 days. OD-, CUS-, and OD + CUS-treated groups and an untreated control group were compared in terms of behavior, endocrine status and brain histology. Results: There were significant differences among the four groups in the behavior tests (*p* < 0.05), especially in the sucrose preference test, where there was a significant decrease in the OD group compared to the control group. ACTH and CORT concentrations were significantly higher in the OD + CUS group than the control group (*p* < 0.05). Expression of GR and 5-HT_2A_R in the PFC, amygdala and hippocampal CA1 was significantly higher in the OD, CUS and OD + CUS groups than the control group (*p* < 0.05). Conclusion: OD promotes depression-like behaviors through peripheral and central pathways via the HPA axis, GR and 5-HT system.

## 1. Introduction

Depression is a common and relapse-prone disorder accompanied by symptoms including loss of pleasure, lack of interest, appetite loss, and even suicidal tendency in serious situations [1,2,3,4]. Many studies have shown that chronic stress can cause neuronal damage and depression; however, the pathogenesis and specific mechanisms underlying depression remain unclear [5]. The chronic unpredictable stress (CUS) model was first described in the 1980s by Katz [6] and developed by Willner and colleagues [7,8,9,10]. The CUS model of depression is considered a prototypical example of depression because it has good predictive validity, face validity and construct validity [8,9,11]. However, the CUS model has not considered factors related to dentistry, e.g., occlusal disharmony (OD).

In clinical dentistry, OD is defined as “a phenomenon in which contacts of opposing occlusal surfaces are not in harmony with other tooth contacts and/or the anatomic and physiologic components of the craniomandibular complex” [12]. Patients suffering from OD may complain about unexplained physical symptoms and mood disorders (e.g., anxiety and depression) [13,14,15]. Moreover, a significant association was established between slight changes in blood flow and prefrontal deactivation during chewing and OD [14]. The prefrontal lobe is involved in emotional modulation. Hara et al. described OD as “a persistent complaint of uncomfortable bite sensation for more than 6 months, which does not correspond to any physical alteration and caused significant functional impairment" [16]. Based on a Japanese patient cohort, Oguchi H et al. suggested applying OD treatment to take into account psychiatric disorders [15].

In animal experiments, OD has previously been established by bite-raising single crowns or by cutting the cusps [17,18]. High anxiety levels, low probing in behavior tests and increased plasma corticosterone were observed, which is a marker of chronic stress [17]. OD has also been shown to enhance the excitability of the lateral habenular nucleus (LHb) in the brain [19], and increase the expression of serotonin 2A receptors (5-HT_2A_R) in the hippocampus [17], which are closely related to the regulation of emotion, and CUS results in depression-like behaviors. OD presents as a chronic stress [17,18], so it is conceivable that OD may also induce depression-like behaviors. However, to the best of the authors’ knowledge, the relationship between OD, CUS and depression has not been studied in animal experiments.

Therefore, the aim of our study was to investigate the effects of OD with accurately controlled attrition of the molars of rats. The potential mechanisms of depression-like behavior effects include hypothalamic–pituitary–adrenal (HPA) axis activity, which results in elevated serum concentrations of corticotropin-releasing hormone (CRH), adrenocorticotrophic hormone (ACTH) and corticosterone (CORT), indicators of depression-like behaviors [20,21,22], and the expression of glucocorticoid receptors (GR) and 5HT_2A_R in the prefrontal cortex (PFC), amygdala and hippocampal CA1 areas [23,24]. We hypothesized that OD is an exacerbating factor for depression-like behaviors.

## 2. Materials and Methods

### 2.1. Animal Preparation

All experimental procedures were approved by the Animal Care and Use Committee of Fujian Medical University (Fujian, China) in accordance with the Guide for the Care and Use of Laboratory Animals of the National Institutes of Health (No: LA2013-76). Twenty-four male Sprague–Dawley rats weighing 160–180 g were used (Vital River Laboratory Animal Technology Co., Ltd., Shanghai, China). All rats were maintained in controlled temperature (22 ± 1 °C) and humidity (50 ± 5%) and a 12 h light/dark cycle (lights on at 8:00 a.m.) with food and water available ad libitum.

### 2.2. Anatomy of Molar Teeth in Sprague–Dawley Rats

Rats were decapitated, and molar samples on the left and right sides of the maxilla were scanned with a μCT100 scanner (SCANCO Medical AG, Wangen-Brüttisellen, Switzerland). Ground sections of the teeth in the coronal and sagittal planes were made (Figure 1a) based on the images of the molars, where the mean thickness of enamel and dentin in Sprague–Dawley rats was determined to be 1 mm.

### 2.3. Animal Model of Occlusal Disharmony (OD)

Individual trays for the rats were prefabricated with lightly-cured resin materials (Figure 1b). The rats were anesthetized by ether narcotization. Impressions of the maxillary dentition were made with silicone rubber (Dental Milestones Guaranteed, Hamburg, Germany) and then poured into a cast model. The model was scanned with an inEos X5 (Dentsply Sirona, Charlotte, NC, USA) scanner, and the data were imported into 3D software. According to the anatomical morphology of rat molars, we used the above software to prepare for the removal of 0.5 mm of the cusp on one side and 0.25 mm on the other side and designed a 3D-printed target restoration space guide (TRS guide). In the OD + CUS group and the OD group, the TRS guide was used, and the maxillary molars on both sides were removed to obtain a precisely controlled rat model of OD with uneven abrasion and unbalanced vertical distance on both sides. The model was scanned and compared to the original model to determine the accuracy.

### 2.4. Animal Model of Chronic Unpredictable Stress (CUS)

All stressors were randomly scheduled and altered every day to maintain an unpredictable procedure.

### 2.5. Experimental Design

The rats were randomly divided into 4 groups: the CON group (rats were anesthetized to keep their mouth open for 5 min; *n* = 6), OD group (rats were anesthetized and subjected to OD only; *n* = 6), CUS group (rats were anesthetized to keep mouth open for 5 min and subsequently subjected to chronic unpredictable stress; *n* = 6), and OD + CUS group (rats were anesthetized and subjected to both OD and chronic unpredictable stress; *n* = 6). Starting on day 8, the rats associated with OD groups were subjected to experimental occlusal disharmony. From day 21 to day 56, the rats associated with CUS were subjected to 8 various stressors. The experimental procedure is illustrated in Figure 2a.

### 2.6. Behavioral Tests

#### 2.6.1. Weight Measurement

The body weight of the rats was measured every week, and body weight increases were recorded and compared among the four groups.

#### 2.6.2. Elevated Plus Maze (EPM) Test

The EPM apparatus was made of black plexiglas and elevated 70 cm from the floor. It consisted of two open arms (50 cm × 10 cm each) and two equally sized closed arms with 40 cm walls. The four arms were connected by a central zone (10 cm × 10 cm) in a shape of a plus sign. The tests were conducted on days 7, 21 and 56 between 10:00 a.m. and 12:00 a.m. under dim light conditions. All rats were individually placed in the central zone of the maze facing a closed arm and allowed to freely explore the EPM for 5 min. The following parameters were recorded by two independent observers who were blinded to the group assignment and sat quietly 2.5 m away from the maze: (1) frequency of open arm entries (OE); (2) frequency of closed arm entries (CE); (3) time spent in open arms (OT); (4) time spent in closed arms (CT). Entry into an arm was defined as the rat placing its four paws in one arm of the maze.

#### 2.6.3. Open-Field Test (OFT)

The OFT was used to evaluate both locomotor and exploratory activities in all groups. The open-field apparatus was a wooden box (100 cm × 100 cm × 50 cm) divided into 9 equally sized squares, with a vertical video camera placed 1 m above the test field. Each rat was gently placed in the center of the enclosure and allowed to freely explore for 5 min, and the number of squares that a rat crossed and the number of times that a rat raised its paw was videotaped to assess horizontal and vertical activities.

#### 2.6.4. Sucrose Preference Test

Adaptation training for drinking a sucrose solution in rats requires 12 h. During the test period, the sucrose water bottle and the drinking water bottle were swapped to avoid the rat’s preference for the location. Sucrose preference was calculated using the following formula (1):(1)$$\mathrm{Sucrose}\;\mathrm{preference}\;\left(\%\right)=\frac{\mathrm{Sucrose}\;\mathrm{water}\;\mathrm{intake}}{\mathrm{Total}\;\mathrm{fluid}\;\mathrm{intake}}\times 100\%$$

### 2.7. Enzyme-Linked Immunosorbent Assay (ELISA) Analysis

All rats were anesthetized in a random order with an overdose of sodium pentobarbital (40 mg/kg). A capillary tube was dipped in sodium heparin solution, some liquid was aspirated, the capillary tube was then turned upside down and washed, and then it was vertically inserted into the back of the rat’s eye socket. Once the blood was smoothly flowing, it was quickly collected into EP tubes, and serum was extracted. Serum concentrations of CRH, ACTH and CORT were measured with ELISA kits (Enzo Life Science Inc., Farmingdale, NY, USA) in accordance with the manufacturer’s protocols. The optical density was measured at 405 nm using a microplate reader (MR96A; Mindray Co., Shenzhen, China).

### 2.8. Western Blot of GR and 5-HT_2A_R Expression

Protein was extracted with a whole cell lysis assay (Beyotime, Shanghai, China), and the concentration was evaluated by an enhanced BCA protein assay kit (Beyotime, Shanghai, China). Equal amounts of protein samples were separated by sodium dodecyl sulfate–polyacrylamide gel electrophoresis (Bio-Rad Laboratories, Hercules, CA, USA) and transferred onto a polyvinylidene fluoride membrane (Millipore, Burlington, MA, USA). Following incubation with specific antibodies, we probed proteins with an enhanced chemiluminescence kit (Bio–Rad Laboratories, Hercules, CA, USA). Bands were quantified by ImageJ software. Specific antibodies were used to detect GR (1:1000, Abcam, Cambridge, UK), 5-HT_2A_R (1:500, Sigma-Aldrich, St. Louis, MI, USA) and GAPDH (1:5000, Proteintech Group, Wuhan, China).

### 2.9. Immunohistochemistry of GR and 5-HT_2A_R Expression

Brain slices of the PFC, amygdala and hippocampus were immunostained for GR, 5-HT and 5-HT_2A_R expression using a three-step method. Following antigen retrieval, the tissue was washed in 0.1 mol/L phosphate-buffered saline (PBS) (3 × 5 min). The slices were sealed with goat serum and incubated overnight at 4 °C with primary rabbit anti-GR polyclonal antibody (1:100 dilution, #12041s, CST, Boston, MA, USA), rabbit anti-5-HT polyclonal antibody (1:50 dilution, ab6336, Abcam, Cambridge, UK) or primary rabbit anti-5-HT_2A_R polyclonal antibody (1:200 dilution, ab16028, Abcam, Cambridge, UK). After universal resistance to biotin labelling was added, the sections were incubated with horseradish peroxidase-labelled streptomycin. Immunoactivity was revealed with 3,3′-diaminobenzidine (DAB, Beijing Zhong Shan-Golden Bridge Biological Technology Co., Ltd., Beijing, China) under a microscope and stopped with distilled water. DAB counterstaining was performed with hematoxylin. Finally, slices were gradually dehydrated with 80%, 95%, and 100% ethanol twice, cleared with xylene three times and adhered to glass slides. Quantitative imaging was performed by a blinded method. Six random fields (400×) in each coronal section were selected, and a total of 5 sections per sample were obtained to capture cells throughout the PFC, amygdala and hippocampus. The integral optical density (IOD) of GR and 5-HT_2A_R immune-positive signals calculated by Image-Pro Plus 6.0 was used as a quantitative standard for GR and 5-HT_2A_R expression, and the mean value of each sample was used for analysis.

### 2.10. Statistical Analysis

Experimental data analyses were performed with a one-way analysis of variance (ANOVA) across the groups (SPSS 20.0: SPSS Inc., Chicago, IL, USA). Student–Newman–Keuls (SNK) tests were subsequently used for post hoc comparisons. All data are presented as the mean ± SEM, and *p <* 0.05 was considered to be statistically significant.

## 3. Results

### 3.1. Weight

No significant differences appeared in the weight on day 7 and 21. However on day 56, there was a significant reduction in the OD + CUS group, CUS group and OD group compared to the CON group (*p* < 0.05), whereas there were no significant differences between the CUS group and the OD group (Figure 2c). 

### 3.2. Sucrose Preference Test

No significant differences appeared in SPT on day 7 (Figure 2d). However on day 21, there was a significant reduction in the OD group compared to the CON group (*p* < 0.05), and on day 56, there was a significant reduction in the OD group compared to the OD + CUS group, CUS group and CON group (*p* < 0.05). Sucrose preferences in the CUS and OD + CUS groups were lower than the CON group (*p* < 0.05), but there was no significant difference between the CUS group and the OD + CUS group (Figure 2d).

### 3.3. Open-Field Test

The OFT was used to explore motivated and anxiety-like behaviors in a novel environment. No significant differences appeared in the horizontal scores on day 7 or the vertical scores on day 7 and 21. There was a significant reduction in the horizontal scores for the rats in the OD + CUS group compared to the CON group (*p* < 0.05) on day 21, and significant reductions in the OD group, CUS group and OD + CUS group compared to the CON group (*p* < 0.05) on day 56. The horizontal scores of the CUS group were lower than OD group (*p* < 0.05), and there were no significant difference between the OD group and the OD + CUS group. The vertical scores for the OD + CUS group were greater than the CON group (*p* < 0.05) on day 56, and there were no significant differences between the OD group, CUS group and CON group. (Figure 2e)

### 3.4. Elevated Plus Maze Test

Anxiety levels of the rats were measured by the EPM. There were no significant differences among the groups for open arm entries (OE, *p* > 0.05), time spent in open arms (OT, *p* > 0.05), open arm entries (OE, *p* > 0.05), time spent in closed arms (CT, *p* > 0.05) and closed arm entries (CE, *p* > 0.05) on day 7. On day 21, CE and OE were significantly decreased in the OD + CUS group compared to the CON group (*p* < 0.05), and CT was significantly decreased in the OD + CUS group compared to the CUS group (*p* < 0.05). On day 56, CT was significantly increased in the OD + CUS group and OD group compared to the CON group (*p* < 0.05); OT showed a decreasing trend in the OD + CUS group and OD group, but there were no significant differences between the groups; CE showed an increasing trend in the OD + CUS group, but there were no significant differences between the groups; and OE was significantly decreased in the OD + CUS group compared to the CON group (*p* < 0.05) (Figure 2f).

### 3.5. Serum Concentrations of CRH, ACTH and CORT

To demonstrate depression in the experimental rats, the serum concentrations of CRH, ACTH and CORT were tested (Figure 2g–i). There were no significant differences among the groups for serum CRH (*p* > 0.05) but the OD + CUS group had significantly higher serum ACTH and CORT levels compared with CON rats, and significantly higher ACTH levels compared with the OD group (*p* < 0.05) (Figure 2g–i).

### 3.6. Western Blot for GR and 5-HT_2A_R Expression

A significant decrease in GR immunoreactivity was detected in hippocampal CA1 and PFC samples from the OD + CUS group compared to the CON group. The levels of 5-HT_2A_R protein were also increased in hippocampal CA1 and PFC samples from the OD + CUS group (*p* < 0.05) compared to the CON group (Figure 3a,c). Similarly, 5-HT_2A_R protein levels were significantly increased in the hippocampal CA1 of the OD group and in the PFC of the CUS group (*p* < 0.05) compared to the CON group (Figure 3b,d).

### 3.7. Immunohistochemistry of GR and 5-HT_2A_R Expression

Immunohistochemical staining for GR and 5-HT_2A_R were conducted in the hippocampal CA1, amygdala and PFC. GR positive staining in the hippocampal CA1 and PFC areas of the CON group was stronger than the OD, CUS and OD + CUS groups (*p* < 0.05), and GR staining in the hippocampal CA1 of the OD group was stronger than the OD + CUS group. The opposite trend was observed in the amygdala (Figure 4a,c). The results of 5-HT_2A_R staining showed strong positive staining in the hippocampal CA1, amygdala and PFC of the OD + CUS, OD, CUS groups that was significantly stronger than the CON group (*p* < 0.05). In addition, 5-HT_2A_R staining in the hippocampal CA1 of the OD + CUS group was stronger than the CUS group (*p* < 0.05); in the amygdala, staining in the OD + CUS group was stronger than the OD and CUS groups (*p* < 0.05); and in the PFC, staining in the OD + CUS group was stronger than the OD group (*p* < 0.05) (Figure 4b,d).

## 4. Discussion

Depression is a widespread mood disorder that has significant adverse effects on personal health. The main symptoms of depression are decreased responsiveness to sexual rewards and locomotor activity and increases in aggression, weight loss and disrupted sleep patterns. Increasing evidence has suggested that chronic stress plays a key role in inducing depression. OD has been shown to be a chronic stressor and was suggested to cause osteoporosis and cognitive dysfunction [25]. Therefore, it is important to determine whether OD can affect depression levels in patients, including OD patients who are exposed to a variety of stressors. We found that only OD could induce weight loss and sucrose preference, increase CT in the EPM and decrease horizontal scores in the OFT. Based on CUS, OD lowered entries into OAs and elevated ACTH and CORT levels in serum. Significant changes in GR and 5-HT_2A_R expression in the PFC, amygdala and hippocampus were induced by OD in rats, with or without CUS, and the more obvious alterations observed in the OD + CUS group may indicate that OD may be a promoting factor for depression through the hypothalamus–pituitary–adrenal (HPA) axis and 5-HT system.

### 4.1. Design of the Experimental Procedure

According to previous research, 90 min after establishing OD, the expression of CRH mRNA showed an initial increase in the paraventricular nucleus (PVN) in the OD group that was followed by a return to control levels on day 14, indicating that the early stages of the occlusal disorder had a significant impact on the body, and there were adaptations to this condition over time [26]. The CUS model of depression takes a longer time to stress rats; however, once the model develops, depressive behavior lasts for weeks or months [27]. In the present study, the OD model was established on day 7, and the rats adapted to the OD for 14 days. The CUS groups were then subjected to two random stressors per day from day 21 to day 56 (Figure 2a). The procedure replicates clinical situations that dentists face in their actual practice [17] and indicates whether the compensation for OD will become disrupted after exposure to chronic stress. 

### 4.2. Development of a New OD Animal Model

An epidemiological investigation showed that malocclusion contributes to psychological stress in young Japanese adults [28]. OD has also generally been considered as a stressor that promotes anxiety in laboratory rats [17,29]. Cutoff maxillary molar cusps and bite-raising single crowns in rats [17,18,22] are conventional methods of modeling experimental OD with prosthetics. Clinically, the thickness of a prosthesis exceeding 48.4 µm above the original occlusion can result in a patient’s self-perception [30]. These raised bite problems associated with restoration can be adjusted for by dentists. Moreover, the balanced wear of molars is rare in dentistry as a result of differential tooth abrasion on the two sides with dental attrition. Without considering the anatomical structure of the molars and oral protection, vulnerable dental pulp and soft tissue damage may have affected our results. Unlike previously described models, in this study a digital scanning method was employed to build the model, and 3D software was used to calculate tooth wear measurements. Comparison of the vertical distances between the two sides resulted in differences, which better simulated the clinical situation. To control abrasion accuracy, we used individualized TRS guides while protecting oral tissue (Figure 1). The successful establishment of an OD animal model affecting depression was corroborated on days 21 and 56 with the evaluation of body weight, SPF, OFT and EPM behavior results, and circulating CRH, ACTH and CORT levels (Figure 2). The influences on rat behavior were still apparent after 14 days, which is somewhat different from previous studies [26]. This might have been associated with differences in the models, tooth wear or vertical distance variation that can have a deeper effect than simply cutting off cusps. The age selection of the rats might have also had an impact. Masticatory interventions in the early stages of life may increase mental disease susceptibility [31]. This implied that our OD model was potent enough to induce depression-like behavior.

### 4.3. OD Is a Potential Factor Promoting Depression in a Rat Model

An attractive feature of the CUS model is the range of behavior and physiological changes that parallel the symptoms of depression. In particular, CUS causes a generalized decrease in reward sensitivity and an increase in the threshold for brain stimulation reward [8,32], which are core features of depression. Although the CUS model has been criticized for lacking reliability, consistent with previous studies, sucrose intake was significantly decreased in the OD group compared with the CON, CUS and OD + CUS groups in this study, indicating that OD may be better than CUS in affecting reward sensitivity. Furthermore, negative correlations for sucrose intake and body weight in the CUS model were observed; OD-exposed rats exhibited a sucrose preference and weight reduction, which are both depressive-like phenotypes (Figure 2) and characteristic of depression [3,9].

Moreover, anxiety in the EPM has been described in animals subjected to CUS, but it is not commonly observed [8,9]. However, depression may be associated with high levels of anxiety [1]. From day 7 to day 21, there were significant differences in the rate of change in EPM performance (Figure 2f) between the groups, and the variations in tendencies across the groups indicated that the OD-associated groups showed more anxiety than the other groups on day 56. This finding is consistent with previous suggestions that OD is widely considered to cause anxiety-like behaviors [17]. In our study, anxiety could not be only ascribed to CUS, which is consistent with a previous study [8]. CUS and OD additionally affect anxiety-like behaviors. We speculate that OD is an effective factor promoting anxiety (Figure 2e,f). 

### 4.4. Activity of the Hypothalamus–Pituitary–Adrenal (HPA) Axis

Experimental OD in rodents increased plasma CORT levels, which is an indicator of depression and suggests an association between OD and HPA axis activation [25,26]. Chronic or repeated stress, such as OD and psychological stress, increased CRH mRNA in the PVN [26]. Experimental OD, acting as a stressor, was shown to suppress hippocampal choline and histamine levels while elevating catecholamine, 5-HT, nitric oxide (NO) levels and mineralocorticoid receptor (MR) expression, which subsequently reduces hippocampal GR expression [25]. Similar to our study, OD has been shown to cause decreases in GR levels in the PFC and hippocampus and an increase in the amygdala, which resulted to hypersecretion of hypothalamic CRH (Figure 2g and Figure 5a) and arginine vasopressin (AVP), promoted secretion of ACTH (Figure 2g and Figure 5b), and stimulated the synthesis and secretion of CORT [33,34] (Figure 2g and Figure 5c). Sustained stress, such as OD, further enhances the secretion of CORT in rats, disrupts the negative feedback system of the HPA axis, and induces hippocampal damage [25] (Figure 5d). Compared with the CON group, the OD + CUS group increased HPA axis activity significantly, indicated by elevated serum ACTH and CORT compared to the other groups, implying that CUS might enhance the CRH stress response during OD. Moreover, OD induced by abrasion might be a risk factor for hypersensitivity to CUS, which implies that OD promotes the induction of depression-like behavior.

### 4.5. Regulatory Mechanisms of the 5-HT System in Depression

The 5-HT system is assumed to be involved in a wide range of physiological functions and different kinds of normal and disordered behaviors [35]. A primary role for 5-HT has been identified in the mechanisms underlying antidepressant effects [36]. In animal models, long-term selective serotonin reuptake inhibitors (SSRIs) increased intrasynaptic 5-HT levels in the cortex, providing further evidence for the effect of 5-HT [23,37,38]. Although there is no direct correlation between neuronal receptors sensitive to 5-HT, significant differences in the physiological effects of 5-HT receptors have been proved [39,40]. The 5-HT_2A_ receptor has attracted particular attention from neuroscientists since it has been shown to be upregulated in depression and anxiety [41] and to regulate the dopaminergic, glutamatergic, noradrenergic and brain-derived neurotrophic factor (BDNF) systems, all of which are relevant to mood regulation, motivation and cognition [17,42,43]. Additional evidence has indicated that antidepressants bind 5-HT_2A_R with relatively high affinity. One characteristic of these agents is their ability to occupy 5-HT_2A_R in the brain, and it seems to be specific for blocking 5-HT_2A_-mediated responses, successfully reducing CUS-induced depression [23,37,44]. In addition, activity of 5HT_1A_R is related to the underlying mechanisms of depression [39] and contributes to antidepressant action [45]. Arsenic can compete with 5-HT to bind the 5-HT_2A_ and 5-HT_1A_ receptors [45,46,47,48], which weakens aspects of well-being, that is, produces depressant effects. A previous study showed that 5-HT_1A_ receptor-mediated neurotransmission in cortical and limbic areas is enhanced by blocking 5-HT_2A_R [37].

OD in animals further suppresses activation in the hippocampus and the prefrontal cortex, which are also essential for emotion management [49]. Consistent with previous studies, a significant increase in 5-HT_2A_R expression in the hippocampal CA1 and prefrontal cortex area was observed in the OD + CUS group compared with the CON group in our study, which indicated that OD produced changes in mood regulation via the 5-HT system similar to CUS, and when combined, the two induced substantial effects. Additionally, concentrations of plasma CORT were significantly increased only by OD + CUS, which was paralleled by increases in 5HT_2A_R expression in the hippocampal CA1, amygdala and prefrontal cortex areas in present study, which also demonstrates activation of the HPA axis by stress via 5HT_2A_R [50].

### 4.6. Relationships between Depression-like Behavior and OD: Possible Mechanisms Underlying the Effects of OD

Appropriate and adaptive responses to stress involves stimulating neural circuits in structures, such as the hippocampus, prefrontal lobe and amygdala, that communicate with other brain structures, such as the thalamus [35]. In the present study, rats could not maintain complex dynamic homeostasis during exposure to OD, which was demonstrated by decreases in sucrose preference and body weight, and the induction of other typical depressive phenotypes. Stress-responsive systems involve the HPA axis, GR expression and 5-HT system changes in the hippocampus, prefrontal lobe and amygdala. We hypothesize a possible mechanistic account as follows.

OD is always accompanied by tooth loss, tooth wear and aberrant dental occlusion that leads to masseter hyperactivity, which are conducted to the limbic systems through the trigeminal sensory system [17,19]. Therefore, OD may influence the maintenance of sufficient attention and regulate emotion (Figure 6a). In addition, OD modulates the hippocampus via HPA axis activation, and thus, a significant increase in circulating CORT. Chronic and sustained exposure to CORT have detrimental effects that decrease GR expression in the hippocampus and PFC, which is crucial for limiting HPA axis activity [25,51] (Figure 6c). GR modifies the function of the 5-HT system, which can be regarded as part of an integrated structure with the forebrain that is involved in regulating the effects of stress on the hippocampus, PFC, amygdala and hypothalamus [35]. This structure can also regulate the densities of GRs and MRs, controlling the output of CORT and participating in the negative feedback of the HPA axis [52,53] (Figure 6b–d). In the present study, increased 5-HT_2A_ receptor expression (Figure 3 and Figure 4) and decreased density of GRs (Figure 3 and Figure 4) in the hippocampus and prefrontal cortex in the OD + CUS group decreased afferent input for the hypothalamic PVN, which also integrates abnormal information obtained from masseter hyperactivity [19,34,54]. Inhibitory effects on CRH and AVP mRNA expression in parvocellular PVN neurons are subsequently disrupted, with consequences that ascend through the hypothalamic median eminence, releasing CRH and AVP and stimulating the release of pituitary ACTH and CORT (Figure 6a–c). Conversely, high CORT affects TPH2, a rate-limiting enzyme for the synthesis of central 5-HT that influences 5-HT neurotransmission in rats [50] (Figure 6d). The HPA axis, GR, and 5-HT system interact with and influence each other to create a triangular loop with OD as the initiating factor and emotional regulation circuits, such as the prefrontal lobe, amygdala, and hypothalamus, as the structural base. OD disrupts synaptogenesis in CA1 pyramidal neurons and could also inhibit PFC metabolism, activation of the amygdala, and produce a loss of balance between the HPA axis, GR, and 5-HT systems (Figure 6) that fails to regulate the depression-like behavior. In this way, OD may play a promoting role in the development of depression.

### 4.7. Limitations

In our study, the closed test for depression-like behavior was the sucrose preference test, but it would be appropriate to use different tests, such as the tail suspension test and light/dark box test, to measure depressive-like behavior in future studies.

## 5. Conclusions

Overall, our data showed that OD significantly affected weight and SPT behaviors, and when combined with CUS, significantly increased the concentrations of ACTH and CORT and impaired the GR and 5-HT system expression, which mediated depression-like behaviors. Therefore, OD may represent a factor that can promote depression in patients who are experiencing emotional stress, a common situation in clinics.

## Figures and Tables

**Figure 1 brainsci-12-00747-f001:**
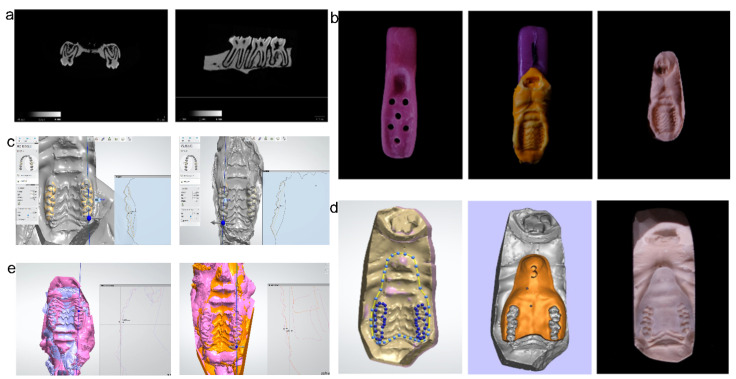
Schematic diagram of experimental occlusal disharmony. (**a**) Three-dimensional reconstructed microtomography images of maxilla, coronal and sagittal sections. (**b**) Individual trays for rats made with lightly-cured resin materials (left), impression of maxillary dentitions (middle) and the cast model (right). (**c**) Virtually prepared teeth in the software. (**d**) Designed and 3D-printed TRS guide. (**e**) Prepared teeth model that was scanned to compare with the original model to assess accuracy.

**Figure 2 brainsci-12-00747-f002:**
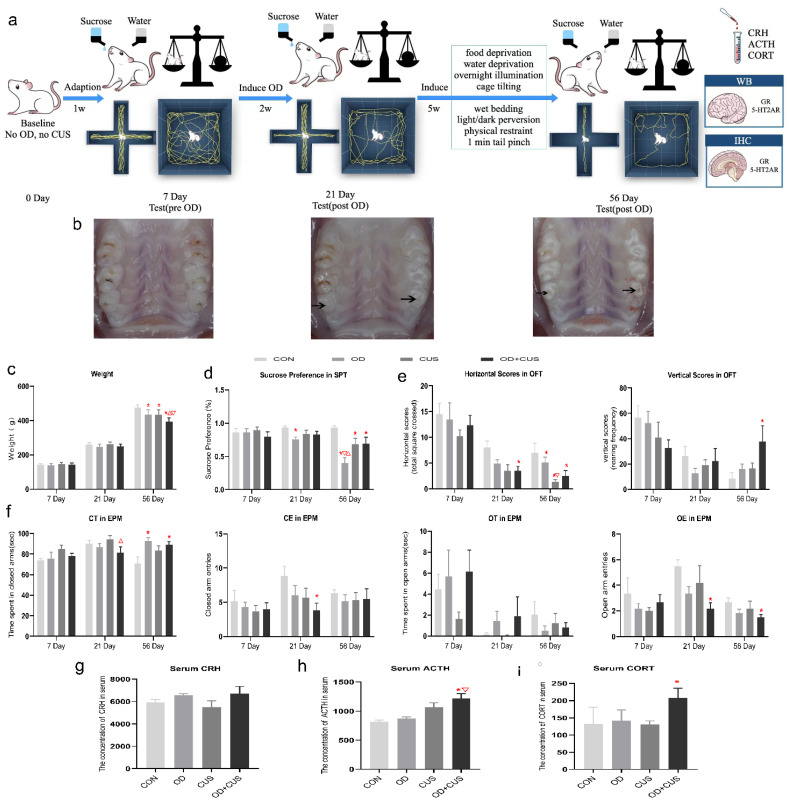
Rats subjected to OD and CUS exhibit depressive-like behaviors. (**a**) Experimental design for the OD and CUS treatments. (**b**) Upper jaw specimens before OD on day 7, and after OD on days 21 and 56. Black arrows indicate the tooth abrasion produced by hand. (**c**) Body weight measurements. (**d**) Sucrose preference test results. (**e**) Open-field test, horizontal scores. (**f**) Time spent in the closed arms of the EPM, number of entries into closed arms of the EPM, time spent in open arms of the EPM, number of entries into open arms of the EPM. (**g**) Serum CRH levels. (**h**) Serum ACTH levels. (**i**) Serum CORT levels. Values represent the mean ± SEM. * *p* < 0.05, compared with the CON group. ▽ *p* < 0.05, compared with the OD group. △ *p* < 0.05, compared with the CUS group.

**Figure 3 brainsci-12-00747-f003:**
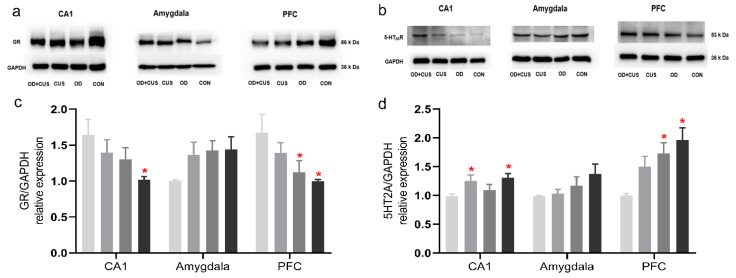
Analysis of GR and 5-HT_2A_R expression in the hippocampal CA1, amygdala and PFC on day 56. (**a**) Western blot for GR protein expression in the OD + CUS, CUS, OD and CON groups. (**a**) Western blot for 5-HT_2A_R protein expression in the OD + CUS, CUS, OD and CON groups. (**c**) Statistical comparison of the blotting data for GR protein levels. (**d**) Statistical comparison of the blotting data for 5-HT_2A_R protein levels. Data are shown as the mean ± SEM. * *p* < 0.05, compared with the CON group.

**Figure 4 brainsci-12-00747-f004:**
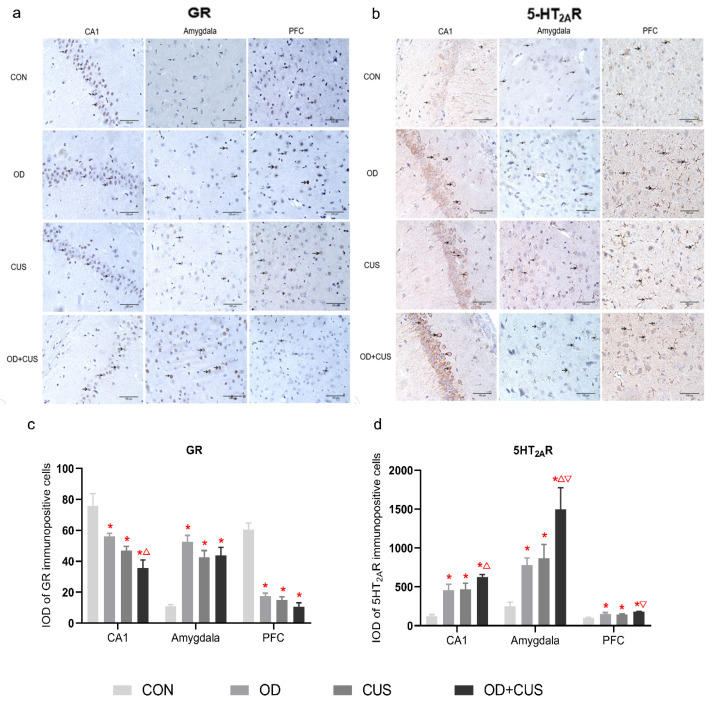
Analysis of GR and 5-HT_2A_R expression in the hippocampal CA1, amygdala and PFC on day 56. (**a**) Representative photographs of immunohistochemical staining from the OD + CUS, CUS, OD and CON groups are shown. (**b**) Representative photographs of immunohistochemical staining for each group are shown. (**c**) The integral optical density (IOD) of GR immune-positive cells. (**d**) The integral optical density (IOD) of immune-positive cells. Data are shown as the mean ± SEM. * *p* < 0.05, compared with the CON group; ▽ *p* < 0.05, compared with the OD group; △ *p* < 0.05, compared with the CUS group.

**Figure 5 brainsci-12-00747-f005:**
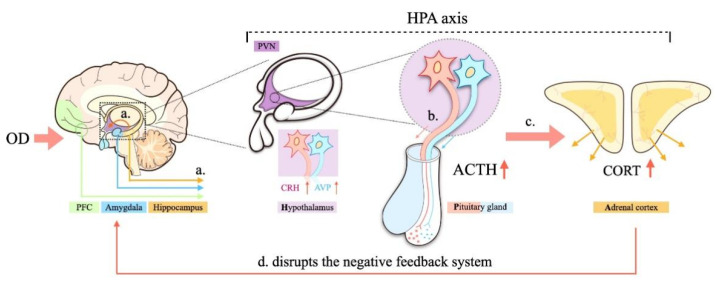
Simplified diagram showing the relationship between OD and the HPA axis. (**a**) OD results in hypersecretion of hypothalamic CRH; (**b**) CRH promotes secretion of ACTH; (**c**) ACTH stimulates the synthesis and secretion of CORT; (**d**) hypersecretion of CORT disrupts the negative feedback system of the HPA axis.

**Figure 6 brainsci-12-00747-f006:**
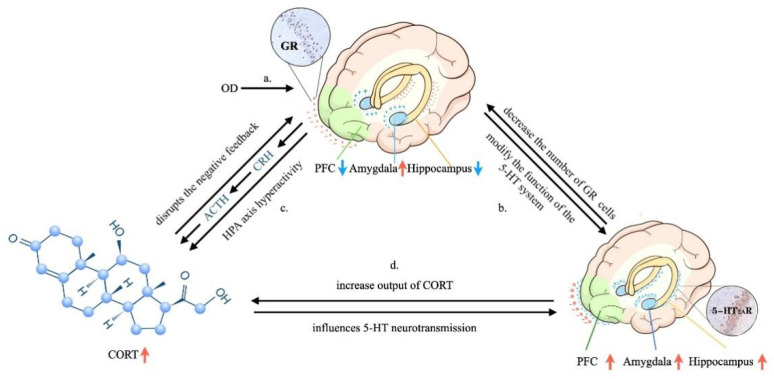
Possible mechanisms underlying the effects of OD. (**a**) OD may influence the regulation of emotion; (**b**) the relationship between the GR and 5-HT systems; (**c**) the relationships among GR, CORT and HPA axis; (**d**) the relationship between CORT and the 5-HT system. ↑ GR, 5-HT_2A_R and CORT levels increased, ↓ GR level decreased.

## Data Availability

The data that support the findings of this study are available from the corresponding author, [author initials], upon reasonable request.

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
