# Peer review of "Occlusal Disharmony—A Potential Factor Promoting Depression in a Rat Model"

_brainsci, 2022, doi:10.3390/brainsci12060747_

Round 1
Reviewer 1 Report
Zhang et al studied some physiological and neuronal parameters after exposure of rats to a combination of chronic mild stress and prior occlusal disharmony (OD), which they present as a novel model of depression. However, the manuscript has severe conceptual gaps and misleading statements:
- Most importantly, as a novel model of depression, behavioural data demonstrating increased depression-like behaviour are essential. However, in the only test for depression-like behaviour performed, the sucrose preference test, “the sugar water intake of rats experiencing OD was lower than that of other groups (Fig. 2d)” i.e. MD and OD did not induce substantial depression like behaviour. This disqualifies the model. Most importantly, it is needed to distinguish between sickness behaviour and depression-like behaviour.
- A better description of the CMS model is needed: How long was the CMS? In the abstract, the authors state 28 days, whereas in the material and method section they state 35 days? Also, which stressors were performed during CMS?
- According to the reference, the stressors applied were not mild at all, as 24 hours foot and water deprivation, are heavy stressors for rats. Please change in all parts of the entire manuscript. Also, please use consistently the description of the model, i.e. either chronic mild stress or chronic unpredictable stress.
- Please give the references of all your discussed statements, e.g.:
- 4.1 depressive-like behavior following CMS lasts up to 3 months.
- 4.1 The procedure replicates clinical situations that dentists face.
- 4.3 Experimental OD, acting as a stressor, was shown to suppress hippocampal choline and histamine levels while elevating catecholamine, 5-HT, nitric oxide (NO) levels and mineralocorticoid receptor (MR) expression, which subsequently reduces hippocampal GR expression.
- The authors discuss increased ACTH levels in OD+CMS rats, which was not shown in the present manuscript (4.3). In the conclusion, the authors also refer to increased CRH expression, which was not shown in the present study.
- Depression is especially high in females. Did the authors include female rats? Please give a statement on sex differences within your findings.
- The description of the SPT in the results and in the discussion section (4.2.2) is contradictory to the results shown in the graph. CMS alone already reduced sucrose preference. The combination of CD+CMS did not additionally affect depressive-like behavior, as OD+CMS rats even preferred sucrose compared to OD rats. Therefore, it seems that CMS following OD rescues sucrose preference and increases it to the level of CMS rats compared to OD rats. Therefore, I do not agree on their statement of OD+CMS as an animal model of depressive-like behavior.
- In the discussion (4.2.2), the authors mention a much more severe increase in anxiety in OD treated rats. Despite increased rearing in the OF, OD only increased the time spent in the closed arms on the EPM, but not open arms. Thus, their statement on anxiety-like behavior should be formulated more careful. Moreover, the authors link their statement to Figure 2c, showing body weight, not anxiety-like behavior on the EPM.
- Might OD maladaptations be induced by pain? Did the authors measure the total amount of water and food consumption? Since the authors stated that OD animals have problems with eating, chronic pain might be an underlying aftereffect, affecting the measures parameters, and reducing sucrose preference. Additionally, in the present study, it would be suitable to use a different test to measure depressive-like behavior that does not involve drinking or eating.
Reviewer 2 Report
This is an interesting study that aims to assess the effect of OD on depression.
Minor issues:
Figure 1c is not clear
figure 2 . The authors say "Upper jaw specimens after OD on days 7, 21 and 56. Black arrows indicate the tooth abrasion produced by hand.". However no black arrows appear in the figure.
figure 3 The scale bar is very small and cannot be seen
figure 3d. I wonder if the authors can add arrows to help the reader identify the cells. Also the number of the cells in the OD+CMS seems equal to control. The same comment can be applied to figure 3e. Overall, the figures quality in general could require further improvement.
Also I wonder if the authors could include the methods of counting the cells for figure 3d. Major issue.
The manuscript aims to assess the effect of OD on depression. However, as the authors use CMS and CMS+OD groups, the reader is a little bit lost in the findings as it is not easy to understand what is the proven effect of OD on depression and whether it is based on CMS or not. I think that adding a paragraph to the discussion section can help the reader understand the significance of the results.
Reviewer 3 Report
Comment
- It is possible that most of the readers are not familiar to occlusal disharmony (OD). The authors need to provide more introduction for the etiologies, types, prevalence and related impairments of OD.
- “It is conceivable that OD can also exacerbate CMS-induced depression-like emotions.” It seemed to be the authors’ hypothesis. However, how the hypothesis came from is unclear. The authors need to provide more introductions for how they came to this hypothesis.
- Tang et al. (2017) (reference 16) found that “the significant changes in exploratory behaviors, serum CORT concentration, 5-HT and 5-HT2AR expressions induced by OD in rats with or without chronic psychological stress (PS), and more obvious alterations in rats with chronic PS.” The result of Tang et al.’s study should be well introduced in Introduction section to be the basis of this study.
- The authors should recheck the ways they cited the references. For example,
- Regarding reference 15 (Oguchi et al., 2017), the authors said “Oguchi H et al suggested applying OD treatment to manage psychiatric disorders.” However, the original conclusion of the study was “OD treatment should take into account the underlying psychiatric disorder manifesting as physical complaints.”
- Furthermore, regarding reference 14 (Narita et al., 2019), the meaning of prefrontal deactivation should be introduced.
- “OD has previously been established by bite-raising single crowns or by cutting the cusps.” The citation should be appeared at the end of the sentences.
- “High anxiety levels and low probing in the behavioural tests and increased plasma corticosterone were observed.” IT needed a citation.
- “OD has also been shown to enhance the excitability of the lateral habenular nucleus (LHb) in the brain, increase the expression of serotonin 2A receptors (5-HT2AR) in the hippocampus, and promote anxiety like behaviour in rats[16, 17].” The conclusion of increased expression of serotonin 2A receptors (5-HT2AR) in the hippocampus, and promote anxiety like behaviour in rats came from reference 16 (Tang et al., 2017). However, I cannot find where the conclusion of “enhanced excitability of the lateral habenular nucleus (LHb) in the brain” came from. The reference 17 (Ekuni et al., 2011) found that occlusal disharmony increases amyloid-β in the rat hippocampus. The authors need to rechecked the citations.
Round 2
Reviewer 2 Report
I do not have further comments.
Reviewer 3 Report
The authors have revised their manuscript based on the reviewer's suggestions.